# A Flagellin-Adjuvanted Trivalent Mucosal Vaccine Targeting Key Periodontopathic Bacteria

**DOI:** 10.3390/vaccines12070754

**Published:** 2024-07-08

**Authors:** Vandara Loeurng, Sao Puth, Seol Hee Hong, Yun Suhk Lee, Kamalakannan Radhakrishnan, Jeong Tae Koh, Joong-Ki Kook, Joon Haeng Rhee, Shee Eun Lee

**Affiliations:** 1Clinical Vaccine R&D Center, Chonnam National University, Hwasun-gun 58128, Republic of Koreaputhsao55@gmail.com (S.P.);; 2Combinatorial Tumor Immunotherapy MRC, Chonnam National University Medical School, Hwasun-gun 58128, Republic of Korea; 3Department of Microbiology, Chonnam National University Medical School, Hwasun-gun 58128, Republic of Korea; 4National Immunotherapy Innovation Center, Hwasun-gun 58128, Republic of Korea; 5Department of Pharmacology and Dental Therapeutics, School of Dentistry, Chonnam National University, Gwangju 61186, Republic of Korea; 6Korean Collection of Oral Microbiology and Department of Oral Biochemistry, School of Dentistry, Chosun University, Gwangju 61452, Republic of Korea

**Keywords:** periodontitis, trivalent mucosal vaccine, flagellin

## Abstract

Periodontal disease (PD) is caused by microbial dysbiosis and accompanying adverse inflammatory responses. Due to its high incidence and association with various systemic diseases, disease-modifying treatments that modulate dysbiosis serve as promising therapeutic approaches. In this study, to simulate the pathophysiological situation, we established a “temporary ligature plus oral infection model” that incorporates a temporary silk ligature and oral infection with a cocktail of live *Tannerella forsythia* (*Tf*), *Pophyromonas gingivalis* (*Pg*), and *Fusobacterium nucleatum* (*Fn*) in mice and tested the efficacy of a new trivalent mucosal vaccine. It has been reported that *Tf*, a red complex pathogen, amplifies periodontitis severity by interacting with periodontopathic bacteria such as *Pg* and *Fn*. Here, we developed a recombinant mucosal vaccine targeting a surface-associated protein, BspA, of *Tf* by genetically combining truncated BspA with built-in adjuvant flagellin (FlaB). To simultaneously induce *Tf*-, *Pg*-, and *Fn*-specific immune responses, it was formulated as a trivalent mucosal vaccine containing *Tf*-FlaB-tBspA (BtB), *Pg*-Hgp44-FlaB (HB), and *Fn*-FlaB-tFomA (BtA). Intranasal immunization with the trivalent mucosal vaccine (BtB + HB + BtA) prevented alveolar bone loss and gingival proinflammatory cytokine production. Vaccinated mice exhibited significant induction of *Tf-*tBspA-, *Pg-*Hgp44-, and *Fn-*tFomA-specific IgG and IgA responses in the serum and saliva, respectively. The anti-sera and anti-saliva efficiently inhibited epithelial cell invasion by *Tf* and *Pg* and interfered with biofilm formation by *Fn*. The flagellin-adjuvanted trivalent mucosal vaccine offers a novel method for modulating dysbiotic bacteria associated with periodontitis. This approach leverages the adjuvant properties of flagellin to enhance the immune response, aiming to restore a balanced microbial environment and improve periodontal health.

## 1. Introduction

Periodontal disease (PD) is a prevalent oral health condition affecting 20–50% of the global population, imposing a significant economic burden and representing a public health problem [1]. In addition to tooth loss, the correlation between periodontitis and systemic diseases has become evident. Related systemic diseases include, but are not limited to, cancer, cardiovascular disease, diabetes, respiratory tract infections, adverse pregnancy outcomes, Alzheimer’s disease, allergies, and rheumatoid arthritis [2,3]. Considering that the dysbiosis of oral microorganisms is tightly related to PD [4], dysbiosis-modifying treatments rebalancing the oral microbiome are considered promising therapeutic approaches to controlling periodontal conditions and preventing or delaying various systemic diseases associated with PD [5,6]. In this regard, a vaccination strategy targeting dysbiosis-causing bacteria may serve as a physiological means for the long-term control of periodontal disease.

PD is associated with a heterogeneous microbial community within the subgingival plaque, which initiates and sustains inflammation, resulting in the progressive destruction of dental tissue [7,8,9]. Cooperative interaction among PD-related pathobionts, which are adept at evading host immune responses, contributes to the persistence of dysbiotic microbial communities [10]. Therefore, effective strategies to control PD should prioritize the reduction in periodontopathic bacteria and the simultaneous modulation of the host inflammatory response. An effective vaccine should disrupt the synergistic periodontopathic interactions by selectively suppressing keystone dysbiotic microorganisms in the dental plaque [11]. In addition, considering that multiple pathogenic bacteria contribute to the destructive inflammation in periodontitis, a preventive or therapeutic vaccine should aim at multiple targets in the polymicrobial community [12].

Given that periodontal disease manifests within the oral mucosa, efficacious periodontal vaccines should be capable of eliciting protective immune responses in the oral mucosa [13]. Injected vaccines are generally poor inducers of mucosal immunity and are, therefore, less effective against infections at mucosal sites [14]. To induce protective immune responses at mucosal sites, mucosal vaccines eliciting secretory antibodies and cellular immune responses at the portal of entry are being actively developed [15]. Mucosally administered antigens are generally less immunogenic and inclined to induce tolerance. In this context, the induction of mucosal immunity through vaccination is a rather tricky task, and potent mucosal adjuvants, vectors, or other unique delivery systems are required [16]. Bacterial flagellin, a cognate ligand for Toll-like receptor 5 (TLR5) on the cell surface and the NAIP5/NLRC4 inflammasome in the cytosolic compartment, has potent mucosal immune-modulatory activities [17,18]. Flagellin has shown potent adjuvant activities in numerous mucosal vaccine formulations as a component of both mixtures and fusion proteins [18,19,20,21,22].

An appropriate animal model system that reproduces human clinical situations is essential for translating vaccines. Many animal models have been devised in PD studies to imitate human pathophysiology. Experimental periodontal infections have been generated through oral gavage, silk ligatures, ligatures soaked in bacteria, or a combination of these methods [23]. Since the oral microbiome is distinctively different between humans and mice, a very high dose of human periodontopathic bacteria should be gavaged in a sticky formulation into the oral cavity to reproduce human PD in mice. However, a high-dose periodontopathic bacterial inoculation may not be enough to induce human-like PD in mice, depending upon the causative pathogens. Ligature-induced PD models are helpful in vivo experimental systems to explore pathogenic mechanisms and therapeutic efficacy [6,24,25,26]. Gavaged pathobionts will colonize the silk ligature for an extended period, which will more likely reproduce human PD than oral gavage alone [27]. We established a temporary-ligature–oral-gavage experimental PD model using specific-pathogen-free (SPF) mice in the present study. *Porphyromonas gingivalis* (*Pg*), a member of the red bacterial complex, plays a central role in chronic periodontitis [28] and contributes significantly to the pathogenicity of the bacterial community during the progression of the disease [10]. The Gram-negative bacterium *Fusobacterium nucleatum* (*Fn*), classified as the “orange complex”, primarily acts as a bridging colonizer that promotes the transition between early and late colonizers during plaque formation [29,30]. *Tannerella forsythia* (*Tf*), a component of the red complex bacteria, has been observed in a chronic and aggressive form of periodontitis. Moreover, under particular conditions, *Tf* exerts a significant synergistic impact on the severity of periodontal disease [31,32,33,34,35,36,37,38]. Among *Tf*’s virulence factors, BspA is crucial in binding to fibronectin and fibrinogen, causing co-aggregation with various organisms and consequently contributing to alveolar bone loss [39,40]. Previous studies have shown that flagellin potentiates secretory IgA responses against mucosally administered *Pg* Hgp44 and *Fn* FomA antigens [19,22]. For its application in human subjects, its protection against *Pg* and *Fn* would not be enough to cover the wide range of dysbiosis, which drove us to design a clinical-grade anti-PD mucosal vaccine formulation covering the above-mentioned keystone pathobionts in PD. 

In this study, we designed a recombinant mucosal vaccine targeting the surface-associated protein *Tf* BspA by genetically conjugating truncated BspA (tBspA) to built-in adjuvant flagellin (FlaB). We then evaluated the induced protective immune responses of a trivalent vaccine consisting of *Tf*-FlaB-tBspA (BtB), *Pg*-Hgp44-FlaB (HB), and *Fn*-FlaB-tFomA (BtA) in a murine “temporary ligature plus oral infection model”.

## 2. Materials and Methods

### 2.1. Bacteria and Culture Conditions

*Tannerella forsythia* ATCC 43037 (*Tf*) was purchased from the America Type Culture Collection (MA, USA). *Tf* was grown in tryptic soy broth (BD, 211825) supplemented with 0.5% yeast extract (BD, 212750), 0.05% cysteine (Sigma, St Louis, MO, USA, 168149), 10 μg/mL hemin (Sigma, 51280), 5 μg/mL menadione (Sigma, St Louis, MO, M5750), 10 μg/mL N-acetyl muramic acid (Sigma, A3007-100MG), and 1 μg/mL Vitamin K1 (Sigma, Seoul, Republic of Korea, V3501-1G) at 37 °C under anaerobic conditions (85% N_2_, 10% H_2_, and 5% CO_2_) [41]. *Fusobacterium nucleatum subsp*. Polymorphum ATCC 10953 (*Fn*) and *Porhyromonas gingivalis* ATCC 33277 (*Pg*) were grown as described in a previous report [19]. Bacteria were harvested at exponential growth by centrifugation (7000× *g*, 20 min, 4 °C). For live bacterial staining, 10^9^ CFU/mL bacteria were labeled with 10 μM carboxyfluorescein diacetate succinimidyl ester (CFSE) in 1 mL of PBS for 15 min at room temperature (RT). Then, 1 mL of fetal bovine serum (FBS) (Gibco, Waltham, MA, USA, 16000044) was added for 10 min for the blocking step at RT. After two washes with phosphate-buffered saline (PBS), the bacteria were re-suspended in PBS.

### 2.2. Animal and Ethics Statement

All experimental animal procedures were conducted following the guidelines of the Animal Care and Use Committee of Chonnam National University, with the protocol CNU IACUC-H-2022-44. Animal protocols complied with the Animal Welfare Act legislated by the Korean Ministry of Agriculture, Food, and Rural Affairs guidelines.

### 2.3. A Temporary Ligature Plus Oral Infection (LigR + OI) Periodontitis Model Induced by a Combination of a Ligature and Oral Bacterial Gavage

A PD model employing a temporary ligature combined with bacterial gavage was established using SPF BALB/c mice. The ligature was applied by slightly modifying previously published methods [27,42]. Firstly, to suppress the pre-existing oral microbiota, mice were fed with oral antibiotics (2 mg/mL sulfamethoxazole and 0.4 mg/mL trimethoprim) in drinking water for 3 days, followed by 3 days of antibiotic-free recovery [43]. Then, mice were anesthetized and tied with 5-0 surgical silk (FST, Foster, CA, USA, 18020-50) at the maxillary second molar (M2) tooth. Ligatures were applied on day 0, followed by three rounds of bacterial infections (*Tf*, *Pg,* and *Fn*) at 2-day intervals. For the first round of oral infection, mice were orally infected with a mixed suspension of the three bacterial species *Tf, Pg*, and *Fn* (1 × 10^9^ CFU of each bacterium/100 μL) in 2% carboxymethylcellulose (Sigma, SL, USA, 9004-32-4) PBS after ligature placement. On the 6th day after ligature placement, the ligature was removed, and two rounds of oral bacterial infections were conducted at one-day intervals. To evaluate the periodontitis induced in the temporary ligature plus oral infection model, we performed micro-CT and histologic analysis by hematoxylin and eosin staining of the entire right maxilla. 

### 2.4. Intranasal Immunization

Seven-week-old female BALB/c mice were intranasally immunized with vaccine components three times in 2-week intervals under anesthesia, as previously described [19,22]. The vaccines were diluted in PBS to a final volume of 10 μL/nostril (20 μL/mouse), resulting in immunization dosages of 6.1 μg of FlaB-L2-tBspA (BtB), 5.1 μg of BtA, 8 μg of HB, a mixture of 8 μg of HB and 5.1 μg of BtA (HB + BtA), a mixture of 6.1 μg of BtB and 5.1 μg of BtA (BtB + BtA), and the triple combination of 6.1 μg of BtB, 5.1 μg of BtA, and 8 μg of HB (BtB + HB + BtA). 

### 2.5. Micro-Computed Tomography (Micro-CT) Analysis

The right maxilla was excised, fixed overnight in 10% formaldehyde, and then preserved in 70% ethanol at 4 °C until micro-CT scanning. Micro-CT imaging was performed using the Skyscan 1172 CT system (Aartselaar, Belgium). The bone volume fraction (bone volume/tissue volume, BV/TV) was determined following the protocol previously described [19]. A three-dimensional (3D) image from the buccal sides was constructed and analyzed using Mimics software 14.0 (Leuven, Belgium). The linear distance was measured in millimeters (mm) from the cementoenamel junction (CEJ) to the alveolar bone crest (ABC) at four different sites on the buccal side in 3D images [44], including the distobuccal site of the maxillary first molar (M1-2) and second maxillary molar (M2-1), the mesiobuccal of the second maxillary molar (M2-2), and the third maxillary molar (M3). Alveolar bone loss was determined by summing the distances of M2-1, M2-1, M2-2, and M3. 

### 2.6. Hematoxylin and Eosin (H&E) Staining

The maxilla was fixed with 4% paraformaldehyde for 2 days and decalcified in 0.5 M ethylenediaminetetraacetic acid (EDTA) (LPS Solution, Seoul, Republic of Korea, CBE002C) solution in PBS (pH 7.4) for 2 weeks at RT. The decalcified tissue was dehydrated and then embedded in paraffin. Serial sections with a thickness of 5 μm were sliced using a microtome, oriented in a bucco-palatal direction and parallel to this plane [45]. These sections included the distobuccal and palatal roots of the second molar, which were stained using Mayer’s hematoxylin and eosin (Abcam, Camb, UK, ab220365), after which they were observed and scanned through virtual microscopy using the Zeiss Axioscan 7 system (Oberkochen, Germany).

### 2.7. Quantitative RT-PCR (qRT-PCR)

Total RNA was prepared from the gingival tissue to analyze expression levels of periodontitis-related genes, and qRT-PCR was conducted. Mice were intranasally immunized with the trivalent vaccine (BtB + HB + BtA) three times in two-week intervals. Two weeks after the third immunization, mice were treated with ligation plus mixed bacterial infection. One day later, ligatures were removed, followed by an additional bacteria challenge. Gingival tissue was collected 24 h after ligature removal or the last infection. Gingival tissue was excised, and total RNA was extracted using TRizol (Invitrogen, Carlsbad, CA, USA, 15596026). Subsequently, the isolated RNA samples were treated with RNase-free DNase I to eliminate genomic DNA contamination. One microgram of extracted total RNA was used as a template to synthesize cDNA using Topscript^TM^ RT DryMix dT18s (Enzynomics, Daejeon, Republic of Korea, RT200). Quantitative PCR was performed with a Real-Time PCR machine (Applied Biosystem, Foster, CA, USA, A28131) using SYBR Green qPCR PreMIX (Enzynomic, Daejeon, Republic of Korea, RT501M). The qRT-PCR data were generated from the cycle threshold (Ct) values normalized against ribosomal protein L32 expression. Normalized fold change was calculated following the 2^(−ΔΔCt) method. The primer sequences used in the experiment are listed in Appendix A.

### 2.8. Measurement of Antigen-Specific Antibody Titers by Enzyme-Linked Immunosorbent Assay (ELISA)

To assess antigen-specific antibody (Ab) titers, serum and saliva samples were obtained from immunized mice two weeks after the final immunization. ELISA assays were performed following established procedures [19]. The ELISA plates (Corning Laboratories, Kennebunk, ME, USA, 3690) were coated overnight at 4 °C with the antigens tBspA, Hgp44, or tFomA at 1 μg/mL concentration in PBS. Subsequently, plates were washed with sterile distilled water (DW) to eliminate unbound antigens. To block non-specific binding, a blocking buffer [0.5% BSA, 1 mM EDTA in PBST (0.05% Tween-20 in PBS)] was applied at RT for 1 hour. Serially diluted sera or saliva in the blocking buffer was added to the plates and incubated for 2 hours at RT, and then 5 washes were conducted using a microplate washer. Secondary antibodies, HRP-conjugated anti-mouse IgG (Southern Biotech, Birmingham, AL, USA, 103605) or IgA (Southern Biotech, Birmingham, AL, USA 1040-05) antibodies, were used for detection. The signal was developed with the addition of 40 μL of 3,3′5,5′-tetramethylbenzidine (TMB) substrate (BD OptEIA, San Diego, CA, USA, 555214), and the reaction was stopped by adding 40 μL of 1 N H_2_SO_4_. Optical density was measured at 450 nm using a microplate reader. Antibody titers were expressed as the dilution’s reciprocal log2 value, resulting in optical density values at 450 nm that were 2-fold higher than those of blank wells without serum.

### 2.9. Anti-Tf, Anti-Pg, and Anti-Fn Serum Production

To generate anti-*Tf*, mice were immunized with the inactivated *Tf* following procedures described previously [22]. Briefly, 2 × 10^9^ *Tf* cells were inactivated by treatment with 0.3% (*v*/*v*) formalin (T&I, Seoul, Republic of Korea, BPP-9004) overnight. The inactivated *Tf* was thoroughly washed with PBS and mixed with complete Freund’s adjuvant (CFA) (Sigma, St. Louis, MO, USA, CAS9007-81-2) for vaccination. Six-week-old female BALB/c mice were immunized with the inactivated *Tf* cells three times at one-week intervals by subcutaneous injection. Anti-*Pg* and anti-*Fn* were produced following the procedures described in a previous study [19]. The anti-*Tf*, anti-*Pg*, and anti-*Fn* sera were used as positive controls for microscopical observations.

### 2.10. Immunostaining and Confocal Imaging

To ascertain the ability of antibodies to recognize the specific antigens expressed in the live bacteria *Tf*, *Pg*, and *Fn*, immunostaining was conducted using confocal visualization following previously described methods [19].

### 2.11. Cell Culture

KB cells [46] are an epithelial cell line derived from HeLa cells obtained from ATCC (Manassas, VA, USA, CCL-17) and cultured in Eagles’ Minimum Essential Medium (EMEM) supplemented with 10% FBS and ampicillin at 37 °C in aerobic conditions (5% CO_2_). For infection experiments, cells were grown until they reached approximately 90% confluence.

### 2.12. Bacterial Invasion Assay by Flow Cytometry

To investigate the neutralizing efficacy of the anti-sera and anti-saliva, we performed a bacterial invasion assay using flow cytometry, as previously studied [47]. Serum and saliva samples were obtained from mice vaccinated with the respective vaccines or PBS 2 weeks after the final immunization. The KB cells were plated at 1.5 × 10^5^ cells/well in 48-well plates (Costar, Kennebunk, ME, USA, 3548) overnight. A total of 1.5 × 10^7^ CFSE-labeled bacteria were pre-incubated with IgG purified from anti-sera (equivalent to 19 μL of anti-sera/well or 9 μL of anti-sera/well) by using the Melon^TM^ Gel IgG Spin Purification Kit following the manufacturer’s instructions (Thermo Fisher Scientific, Rockford, IL, USA, 45206) or anti-saliva (equivalent to 1/4 or 1/8 dilution) for 1 h at RT. KB monolayers were washed once with serum/antibiotic-free EMEM and subsequently infected at a multiplicity of infection (MOI) of 1:100. The infection was carried out for 4 h at 37 °C under 5% CO_2_. Subsequently, the cell culture medium was aspirated, and the cells were subjected to an additional incubation in a serum-free medium containing gentamicin 300 μg/mL and metronidazole 200 μg/mL for 1 h to kill any remaining extracellular bacteria. The cells were then washed three times with PBS and detached using 0.25% trypsin (Glibo, Canada 25200072). To quench the fluorescence of any remaining extracellular CFSE-labeled bacteria, Trypan blue (Sigma, UK, T8158) was added (0.2%) and analyzed by flow cytometry (BD CantoII, San Jose, CA, USA).

### 2.13. Bacterial Adhesion and Invasion Assay by Confocal Microscopy

To examine bacterial adhesion and invasion by confocal microscopy, KB cells were seeded at a density of 1 × 10^5^ cells/well in 4-well cell culture slides overnight (SPL, Gyeonggi, Republic of Korea, 30104). The cells were then infected with CFSE-labeled bacteria following the protocol mentioned above. After bacterial infection, the medium was aspirated and thoroughly washed with PBS by gentle orbital shaking. The cells were then fixed with 3.2% paraformaldehyde for 10 min at RT, followed by another round of PBS washing. Then, the cells were blocked with PBS containing 1% BSA for 15 min. Afterward, cells were washed with PBS and stained for actin using 1x Rhodonide phalloidin (Molecular probes, OR, USA, R415) for 1 h. Then, nucleic acids were stained with Hoechst 33342 (Invitrogen, Rockford, IL, USA, H3570) at a 1:1000 dilution for 15 min. Following three final washes with PBS, the slide was mounted using DPX Mounting (Sigma, 06522). The images were observed using an LSM510 confocal microscope (Carl Zeiss, Oberkochen, Germany).

### 2.14. Biofilm Inhibition Assay

The inhibitory effect of anti-sera or anti-saliva on *Fn*-induced biofilm formation was determined following established methods [19]. Freshly prepared *Fn* ATCC 10953 cells (1 × 10^9^/well) were incubated with purified IgG from anti-sera (equivalent to 12 μL of anti-sera/well or 6 μL of anti-sera/well) or saliva (equivalent to 1/4 or 1/8 dilution) generated by vaccination with PBS (anti-PBS), a divalent vaccine (anti-BtB + HB), or a trivalent vaccine (anti-BtB + HB + BtA) in high-binding 96-well plates (COSTAR, USA, 3590) at RT for 3 h, with the subsequent addition of *Fn* media at up to 200 μL per well. The plates were further incubated in anaerobic conditions for 24 h, followed by a gentle washing step using PBS. The wells were stained with 25 μL of 0.3% crystal violet for 15 min and then washed gently with PBS. The stained biofilm was extracted by adding 100 μL of 100% ethanol, and the optical density was quantified using a microplate reader at a wavelength of 595 nm.

### 2.15. Statistical Analyses

Unless otherwise stated, the results are presented as the mean ± standard error of the mean (SEM). A Mann–Whitney or an unpaired *t*-test was used to compare the two groups. Statistical analyses were performed using Prism version 8.00 software for Windows (GraphPad Software, San Diego, CA, USA). *p* values < 0.05 were accepted as statistically significant.

## 3. Results

### 3.1. The Establishment of a Temporary Ligature Plus Oral Infection (LigR + OI) Model

To evaluate the efficacy of the periodontal vaccine targeting three major PD-related pathobionts, we looked for the optimal animal model that could testify to the protective efficacy. The model should ensure cooperative pathogenesis by the combination of three PD pathobionts. We thought that a silk ligature would provide the bed for the cooperative infection of multiple pathobionts. First, we tried to combine ligature-induced periodontitis (LIP) with oral bacterial infection (OI) of *Tf*, *Pg*, and *Fn*. The second maxillary molar (M2) of the BALB/c mice was ligated with 5-0 surgical silk. After ligature placement, mice were orally infected with a mixed suspension containing the three bacterial species (*Tf*, *Pg*, and *Fn*). Two more rounds of oral bacterial infections were conducted on the 3rd and 5th days (Appendix A). To assess the extent of periodontitis resulting from mixed ligature–oral infection, a comparison of alveolar bone loss was conducted between two groups, the ligature placement group (Lig) and the mixed ligature–oral infection (Lig + OI) group, by employing micro-CT analysis. Appendix A shows that consistent alveolar bone loss was seen on days 6, 9, 12, and 15 after ligature placement. However, no significant additional alveolar bone loss was observed in the Lig + OI group compared to the Lig group at any time (Lig vs. Lig + OI, *p* > 0.05). These findings imply that the induction of periodontitis via ligature placement (LIP) alone prompts considerable periodontal damage, which suggests that the mouse microbiota could contribute to PD generation. Ligature maintenance was sufficient to generate significant periodontal pathology, rendering it challenging to assess periodontal inflammation and alveolar bone loss attributed to *Tf*, *Pg*, and *Fn*.

Next, we modified the above LIP procedure by temporarily maintaining silk, which would delineate the contribution of *Tf*-, *Pg*-, and *Fn*-mediated inflammation to alveolar bone loss. Briefly, the M2-ligated BALB/c mice were orally infected with mixed bacterial suspensions three times in 2-day intervals. The silk ligature was removed, followed by two more oral bacterial infections on days 6 and 7 (LigR + OI) (Appendix A). To evaluate alveolar bone loss induced by LigR + OI, micro-CT analysis was performed on days 15 and 24. Notably, 9 days (day 15) after ligature removal, LigR + OI mice showed significant alveolar bone loss compared with the LigR group (LigR vs. LigR + OI, *p* < 0.05), demonstrating that the additional two infections further exacerbated the periodontitis induced by the ligature plus three-time oral infections. By day 18 after ligature removal, the LigR group showed recovery of alveolar bone loss, reaching levels similar to naïve mice (naïve vs. LigR, *p* > 0.05) (Appendix A). These findings highlight the relevance of the temporary-ligature–oral infection model (LigR + OI) to specifically assess periodontitis induced by exogenous bacteria at day 15.

### 3.2. The Development of a Tannerella forsythia BspA-Specific Mucosal Vaccine with Flagellin as a Built-In Adjuvant

To develop a vaccine antigen targeting *Tf*, we selected a surface-associated protein, BspA. Computational analysis was then employed to design immunogenic vaccine antigens. We chose the C terminus of BspA (truncated BspA, tBspA) based upon its potential as a B-cell epitope and the likelihood of exposure on the cell surface, as was implemented previously (Appendix A) [19,48]. Subsequently, we engineered a series of eight fusion proteins to generate an optimal built-in adjuvanted mucosal vaccine formulation. These proteins comprise various linker peptides that connect the tBspA antigen and the FlaB adjuvant (Appendix A) [19]. Through pilot studies, we identified FlaB-L2-tBspA (BtB) as a candidate vaccine based on its stability and ability to induce strong TLR5 stimulation (Appendix A). As shown in Appendix A, BtB exhibited superior tBspA-specific serum IgG and saliva IgA responses compared to the mixture formulation of FlaB and tBspA. Conclusively, we developed an optimal built-in adjuvanted anti-*Tf* vaccine candidate that will be co-formulated with the previously reported anti-*Fn*/*Pg* mucosal vaccine [19].

### 3.3. A Trivalent Mucosal Vaccine (BtB + HB + BtA) Prevents Alveolar Bone Loss Induced by a Mixed Bacterial Infection in the Temporary Ligature Plus Oral Infection Model

To evaluate the effectiveness of the trivalent mucosal vaccine in preventing *Tf*-, *Pg*-, and *Fn*-mediated alveolar bone loss, we performed a series of experiments combining intranasal vaccination and a LigR + OI challenge. BALB/c mice were intranasally immunized with a trivalent vaccine formulated with *Tf*-FlaB-tBspA (BtB), *Pg*-Hgp44-FlaB (HB), and *Fn*-FlaB-tFomA (BtA). Two weeks after the last immunization, we challenged vaccinated animals with the temporary-ligature–oral infection regimen (Figure 1a). Nine days after ligature removal, we assessed the effectiveness of induced immune responses by measuring the extent of alveolar bone loss. Mice vaccinated with the trivalent vaccine (Vax + LigR + OI group) were effectively protected from alveolar bone loss caused by an infection with a mixture of *Tf*, *Pg*, and *Fn*. This was evidenced by assessments of CEJ-ABC (LigR + OI vs. Vax + LigR + OI, *p* < 0.05) and the bone volume density (BV/TV) (LigR + OI vs. Vax + LigR + OI, *p* < 0.01) (Figure 1b,c). The vaccine efficacy is discernible in histologic H&E staining (Figure 1d), as shown by the attenuated degradation of periodontal tissue around the second molar. Moreover, vaccination consistently resulted in a significant reduction in myeloperoxidase-positive neutrophil infiltration (Appendix A). These findings demonstrate that the trivalent mucosal vaccine (BtB + HB + BtA) prevents alveolar bone loss caused by mixed PD pathobiont infection.

### 3.4. The Trivalent Mucosal Vaccine (BtB + HB + BtA) Inhibits PD-Related Gene Expression Induced by a Mixed PD Pathobiont Infection

The development and progression of PD are predominantly influenced by the host’s inflammatory responses to periodontal pathogens. The effective suppression of the overgrowth of pathobionts will alleviate inflammatory responses that lead to periodontal tissue damage and alveolar bone loss. To determine whether the trivalent vaccine suppressed periodontal inflammation induced by polymicrobial infection (*Tf* + *Pg* + *Fn*) in the LigR + OI model, we assessed the mRNA expression profile in gingival tissue using qRT-PCR in a separate experimental setting. Initially, we determined the timing of evaluation in the temporary-ligature–oral infection model, specifically targeting gingival tissue inflammation by measuring the transcription of the IL-1β, IL-6, TNF-α, and MMP9 genes. Briefly, mice underwent the ligature procedure, accompanied by oral infections with a mixture of live *Tf*, *Pg*, and *Fn*. Subsequently, the ligatures were removed after one day, followed by an additional bacterial challenge. The expression of PD-related genes was evaluated at 12, 24, and 48 h after ligature removal (Appendix A). Upon comparing the gene expression levels between ligature removal (LigR) and ligature removal combined with oral infection (LigR + OI), notably elevated IL-1β, IL-6, and MMP9 gene expression levels were detected in the LigR + OI group at 24 h following ligature removal (Appendix A). To evaluate the efficacy of the trivalent mucosal vaccine (BtB + HB + BtA) in suppressing inflammation in LigR + OI animals, BALB/c mice were intranasally immunized with the trivalent vaccine (BtB + HB + BtA) and challenged with live bacterial infection following the protocol outlined in Appendix A. Two weeks after the final immunization, the mice underwent a ligature placement/removal procedure along with oral infections. Twenty-four hours after ligature removal, we evaluated gene expression levels in the gingival tissue using qRT-PCR. As shown in Figure 1e, the trivalent mucosal vaccine (BtB + HB + BtA) suppressed the expression of proinflammatory cytokines (IL-1β, IL-6, and TNF-α), a chemokine (CLCX2), extracellular matrix-degrading enzymes (MMP-3 and MMP-9), and RANKL.

### 3.5. Intranasal Immunization with the Trivalent Mucosal Vaccine (BtB + HB + BtA) Induced Antigen-Specific Antibody Responses in Both Mucosal and Systemic Immune Compartments

To substantiate the protective efficacy, we assessed antigen-specific antibodies following intranasal immunization with monovalent (BtB, HB, or BtA), divalent (BtB + HB, BtB + BtA, or HB + BtA), and trivalent (BtB + HB + BtA) vaccines. As shown in Figure 2, each component of the mucosal vaccine induced significant antigen-specific IgG and secretory IgA in serum and saliva, respectively. Immunofluorescence staining for confocal microscopy was conducted to ascertain whether the anti-sera recognized the native forms of the respective antigens (BspA, Hgp44, and FomA) on the surfaces of live *Tf*, *Pg*, and *Fn* cells. In contrast to pre-immune sera, which did not detect antigenic epitopes on the surfaces of all three bacteria, the anti-sera generated by the trivalent vaccine (anti-BtB + HB + BtA) successfully identified the corresponding antigens expressed on the surfaces of *Tf*, *Pg*, and *Fn* (Figure 3). The monovalent (BtB, HB, or BtA) and divalent vaccines (BtB + HB, BtB + BtA) demonstrated similar staining to the trivalent vaccine (Figure 2 and Appendix A). These findings corroborate that the antibodies raised by immunization with the trivalent vaccine specifically recognized infecting pathobionts and subsequently suppressed destructive inflammatory responses in vivo.

### 3.6. Anti-Sera and Anti-Saliva Elicited by Intranasal Immunization with the Trivalent Vaccine (BtB + HB + BtA) Inhibited Host–Bacteria Interactions

Periodontal tissue adhesion and invasion by periodontal pathogens is a crucial step in establishing a destructive infection. Protective antibodies targeting surface-expressed adhesion-related antigens of periodontal pathogens would inhibit the disease’s adhesion/invasion stages [49]. Using flow cytometry and confocal microscopy, we performed invasion assays employing anti-sera and anti-saliva. Purified IgG from anti-sera and anti-saliva were used to evaluate the neutralizing efficacy against the BspA-mediated invasion of epithelial KB cells by *Tf*. Flow cytometry analysis demonstrated that the purified IgG and anti-saliva generated by the trivalent vaccine (BtB + HB + BtA) successfully inhibited *Tf* invasion (Figure 4a,b). The outcome was consistent with the data obtained from confocal microscopy (Figure 4c,d). Additionally, both purified IgG from anti-sera and anti-saliva derived from monovalent (BtB) and divalent (BtB + HB, BtB + BtA) vaccination were also effective in inhibiting *Tf* invasion (Appendix A). In contrast, non-specific antibodies in serum and saliva (anti-HB + BtA and anti-PBS) did not exert any inhibitory effect (Figure 4). These results indicate that the specific inhibitory action of immune sera or saliva against BspA could effectively inhibit oral epithelial adhesion and invasion by *Tf*.

An invasion assay was also conducted to assess whether the anti-sera and anti-saliva from the trivalent vaccine can neutralize the activities of Hgp44, inhibiting the invasion of KB cells by Pg. Non-specific antibodies (anti-BtB + BtA and anti-PBS) were utilized as negative controls. The outcomes demonstrated a significant reduction in the percentage of *Pg* invasion into KB cells with both IgG purified from anti-sera (Figure 5a,c) and anti-saliva (Figure 5b,d) elicited by BtB + HB + BtA immunization. Conversely, treatment with purified IgG from serum or saliva from BtB + BtA- or PBS-administered animals showed no inhibition. These results suggest that the specific inhibitory effect of immune sera or saliva against Hgp44 also contributed to the suppression of *Pg*-mediated inflammation.

### 3.7. Anti-Sera and Anti-Saliva Elicited by Intranasal Immunization with the Trivalent Vaccine (BtB + HB + BtA) Inhibited F. nucleatum-Mediated Biofilm Formation

*Fn* readily forms co-aggregates with various oral bacteria by mediating FomA [50,51]. We conducted a colorimetric biofilm inhibition assay to determine the functional inhibitory impact of antibodies derived from trivalent vaccination on *Fn*-induced biofilm formation. Antibodies in both sera and saliva elicited by intranasal immunization with the BtB + HB + BtA vaccine exhibited a dose-dependent inhibition of *Fn* biofilm formation (Figure 6a,b). Conversely, these inhibitory effects were absent in the groups treated with non-specific antibodies (anti-BtB + HB, anti-PBS). These findings suggest that antibodies induced in both systemic and mucosal compartments functionally inhibited FomA-mediated biofilm formation by *Fn*.

## 4. Discussion

In this study, we propose a mucosal anti-PD built-in adjuvanted vaccine targeting the biofilm-bridging colonizer *Fn* and the two “red complex pathogens” *Pg* and *Tf* to effectively counteract the founders of severe periodontal diseases [52]. While proving the efficacy of the vaccine, we established an experimental system that can be widely used for PD protection studies, employing a temporary silk ligature and oral gavage with live bacteria. Complex interactions of oral microbiota and pathobionts contribute to dysbiosis, leading to PD [4]. Among large numbers of PD-related pathobionts, the three bacteria mentioned above play pivotal roles in the pathogenesis of PD. *Fn* links early colonizers, such as streptococci, and later-stage colonizers, including the red complex species, such as *Pg*, *Tf*, and *Treponema denticola* [52]. *Fn* and *Tf* develop a synergistic partnership in inducing inflammation under PD conditions through interspecies sensing and metabolite exchange [36,53]. *Fn* scavenges reactive oxygen species in the subgingival plaque, providing a favorable growth environment for strictly anaerobic pathogens such as *Pg* [54]. The abundance of *Tf*, *Pg*, and *Fn* can increase microbial biomass and diversity, which is at the center of periodontal inflammation [55]. We generated a trivalent mucosal vaccine that specifically targets the surface-associated virulence factors of *Tf* (BspA), *Pg* (Hgp44), and *Fn* (FomA) by genetically fusing the built-in mucosal adjuvant FlaB. In a previous study, we observed that the built-in FlaB-adjuvanted mixed HB and BtA vaccine induced excellent protective immune responses in oral secretions without interfering with each other [19]. For clinical application, covering only one red complex pathogen (*Pg*) would not be sufficient to have effects on diverse types of PDs. 

The temporary ligature plus oral infection model combines ligature placement/removal in the maxillary molar of mice and oral gavage with live *Tf*, *Pg*, and *Fn*. The initial simultaneous application of the general ligature placement [24,56] and oral infection [25,26,42,57] resulted in rapid and severe, destructive alveolar bone loss (Appendix A). Given that ligature placement may potentially induce microbial accumulation and microulceration within the sulcular epithelium, facilitating the invasion of connective tissues by the oral microbiome [58,59], the extensive alveolar bone loss observed in the initial trial appeared to be influenced by both the existing host microbiota and orally challenged pathogens (*Tf, Pg,* and *Fn*). Distinguishing between PD caused by the host’s oral microbiota and that caused by key heterologous human pathogens is challenging. Through many rounds of pilot experiments, we found that a temporary ligature plus two additional oral gavages provided a clinical setting that would be optimal for testing the efficacy of vaccination. As shown in Appendix A, 9 days after ligature removal and two more rounds of oral infection, we could distinguish the three-bacteria-induced PD from ligature-mediated PD (LigR vs. LigR + OI, *p* < 0.05). Eighteen days after ligature removal, ligation-mediated alveolar bone loss was recovered to the control mouse level (naive vs. LigR, *p* > 0.05). This finding corresponds to previous reports showing that spontaneous regeneration of alveolar bone can be observed 15 days after ligature removal [60]. Following this temporary ligature plus oral infection model, we could address the trivalent vaccine’s protective efficacy (Figure 1b–d). Considering the close association of *Tf*, *Pg*, and *Fn* with peri-implantitis [61], we believe that this modified ligation and oral infection model could be applied to the study of peri-implantitis or other periodontal conditions that accompany tissue injury plus polymicrobial infection. 

The interplay between immune cells and oral microbial dysbiosis results in pathological alveolar bone resorption due to inflammatory responses to dysbiotic microorganisms [62]. We also investigated whether the trivalent mucosal vaccine could modulate inflammation-related gene expression. Cytokine gene expression profiles in periodontal tissue have been reported to undergo changes during PD initiation [25]. We established separate temporary ligature plus oral infection models to address the expression profiles of PD-related proinflammatory cytokines and genes associated with tissue damage (Appendix A). Our experimental results indicate that the trivalent mucosal vaccine was effective in downregulating the expression of PD-related genes, such as proinflammatory cytokines (IL-1β, IL-6, and TNFα), the matrix-degrading enzyme MMP9, and RANKL, in gingivae treated with a live *Tf, Pg*, and *Fn* mixture (Figure 1e). The role of periodontal immune responses in maintaining alveolar bone homeostasis during periodontitis has been emphasized [63]. Clinical and preclinical studies have demonstrated that microorganisms within dental plaque promote osteoclast formation and subsequent alveolar bone loss by upregulating RANKL [64,65]. Our findings offer further mechanistic insights into the impact of mucosal vaccination on PD-associated alveolar bone loss. 

Mucosal vaccination effectively induces secretory IgA in saliva and IgG in gingival crevicular fluid (GCF), interfering with pathogen attachment and colonization at mucosal sites by producing pathogen-neutralizing antibodies [13,19,66]. In contrast to parenteral vaccines, mucosal vaccines generate more effective protective immune reactions by stimulating secretory IgA responses and cell-mediated immunity within mucosal tissues, which serve as the main entry points for mucosal pathogens [15]. Appropriate mucosal adjuvants are crucial for eliciting optimal immune responses [16]. FlaB has been recognized as a potent adjuvant, especially in formulations fused with vaccine antigens that can enhance antigen-specific immune responses at mucosal sites [18,19,20,21,67]. Targeted suppression of keystone pathogens in the oral microbiota will lead to the resurgence of physiological commensals and contribute to establishing a healthy microbial homeostasis [68,69,70]. Another red complex periodontopathic bacterium, *Tf*, exhibits invasiveness toward oral epithelial cells and engages in intricate interactions with other bacteria to synergistically contribute to PD [71]. BspA of *Tf* plays a crucial role in host adhesion and invasion [35]. In this study, we developed an optimal vaccine antigen (tBspA) targeting *Tf* BspA by computational analysis and fused it with FlaB using different linkers (Appendix A). tBspA did not contain leucine-rich repeat domains and was predicted to harbor several B-cell epitopes by BepiPred-2.0 [72]. The construct showing the strongest TLR5-stimulating activity and the best stability was selected for further studies (Appendix A). Positioned as a keystone pathogen, *Pg* can subvert the host’s immune response and drive dysbiosis, even with 10–15% of subgingival plaque [73]. *Fn* promotes the formation and maturation of dental plaque [74,75]. Clinically relevant findings also showed a strong association between polymicrobial biofilms involving *Tf*, *Pg*, and *Fn* causing PD [30,34,53,76,77]. In this study, the flagellin-adjuvanted FlaB-tBspA (BtB) fusion protein was combined with previously reported Hgp44-FlaB (HB) and FlaB-tFomA (BtA) [19] to generate a trivalent vaccine formulation. Mice intranasally administered the trivalent mucosal vaccine (BtB + HB + BtA) elicited antigen-specific serum IgG and saliva secretory IgA responses to each antigen independently, without cross-reactivity. The trivalent vaccine formulation resulted in similar levels of antigen-specific antibody responses to those produced by a mono- or divalent vaccine (Figure 2). In addition, the ability of vaccine-induced antibodies to recognize the native forms of antigens on live bacteria further strengthens the evidence of their potential to neutralize and inhibit the activities of these pathogens (Figure 3). These antibodies demonstrated functional efficacy by inhibiting epithelial cell adhesion and invasion by *Tf* (Figure 4) and *Pg* (Figure 5) and interfering with biofilm formation by *Fn* (Figure 6). The *Fn*-biofilm inhibition finding was further supported by data showing that intranasal vaccination with the trivalent vaccine significantly inhibited *Fn* colonization in gingival tissue (Appendix A). However, we could not observe detectable colonization by *Tf* or *Pg* in the mouse oral cavity in the temporary ligature plus oral infection model, which explains why a temporary ligature was required to elicit osteoclastic inflammation. This limitation also arises from the unpredictable interactions of human oral bacteria *Tf*, *Pg*, and *Fn* with the endogenous mouse microbiome in a ligature model [78]. While murine models are indispensable for preliminary assessment, further validation in larger animal models and eventual clinical trials is required. In conclusion, these findings provide a potential management strategy for PD by specifically targeting key periodontopathic bacteria through immunological approaches. The flagellin-adjuvanted trivalent mucosal vaccine represents an innovative strategy for addressing dysbiotic bacteria linked to periodontitis. By utilizing the immune-boosting effects of flagellin, this vaccine aims to re-establish a healthy microbial balance and enhance periodontal health.

## Figures and Tables

**Figure 1 vaccines-12-00754-f001:**
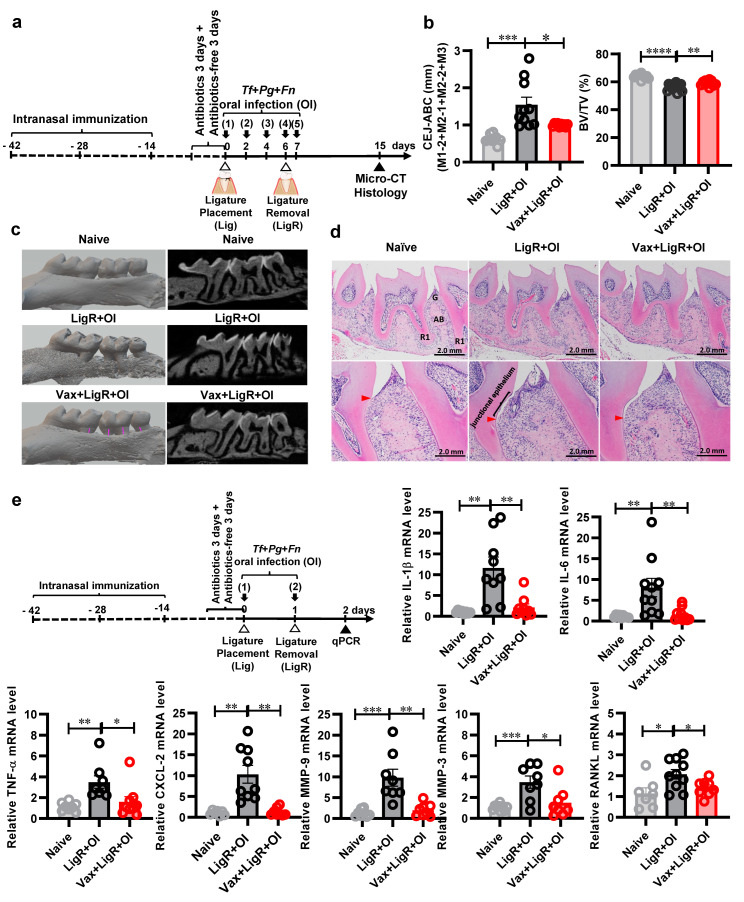
Intranasal immunization with the trivalent vaccine alleviates periodontitis caused by a mixed *Tf*, *Pg*, and *Fn* infection in a temporary ligature plus oral infection model. (**a**) Immunization and experimental schedule. Seven-week-old female BALB/c mice (*n* ≤ 13) were intranasally immunized with PBS or a mixture of 6.2 μg of BtB, 8 μg of HB, and 5.1 μg of BtA as the trivalent vaccine (BtB + HB + BtA) three times at 2-week intervals. Two weeks after the final immunization, a ligature was tied to the second molar of the mouse, which was subsequently orally infected with a mixture of *Tf*, *Pg*, and *Fn* (1 × 10^9^ CFU of each in 100 μL/mouse) three times at 2-day intervals. Two days after the third bacterial infection, the ligature was removed, followed by two additional rounds of bacterial infections daily. Eight days (day 15) after the fifth bacterial infection, mice were sacrificed, and micro-CT and hematoxylin and eosin (H&E) staining were performed (R1, the root of the first molar; R2, the root of the second molar; B, alveolar bone; G, gingival epithelium). (**b**–**d**) The trivalent vaccine protects against alveolar bone loss in mice. (**b**) Measurements of the distance from the cementoenamel junction to the alveolar bone crest (CEJ-ABC) and the bone volume density (BV/TV). (**c**) Representative images of sagittal 3-dimensional and bi-dimensional alveolar bone. The pink line indicates the CEJ-ABC distance on the buccal site. (**d**) H&E stains of periodontal tissue damage. Red arrows indicate the distance of the junctional epithelium. (**e**) The trivalent vaccine inhibits bacterial-induced inflammation in the gingival tissue of mice. The presented schedule illustrates the assessment of mRNA gene expression patterns by qRT-PCR. Similarly, BALB/c mice (*n* ≤ 10) were intranasally immunized with PBS or the trivalent vaccine (BtB + HB + BtA), repeated 3 times at 2-week intervals. After the final immunization, ligatures were placed on day 0, followed by oral infection with mixed *Tf*, *Pg*, and *Fn* (1 × 10^9^ CFU of each in 100 μL/mouse). The ligatures were removed a day later, followed by another round of oral infection. The mRNA gene expression patterns were assessed 1 day after the last oral infection. Naïve (control mice), LigR + OI (mice administered PBS before ligature placement/removal plus oral bacterial infection), and Vax + LigR + OI [mice vaccinated with the trivalent vaccine (BtB + HB + BtA) before ligature placement/removal plus oral bacterial infection]. The results are presented as the mean ± SEM in each group. * *p* < 0.05, ** *p* < 0.01, *** *p* < 0.001, **** *p* < 0.0001.

**Figure 2 vaccines-12-00754-f002:**
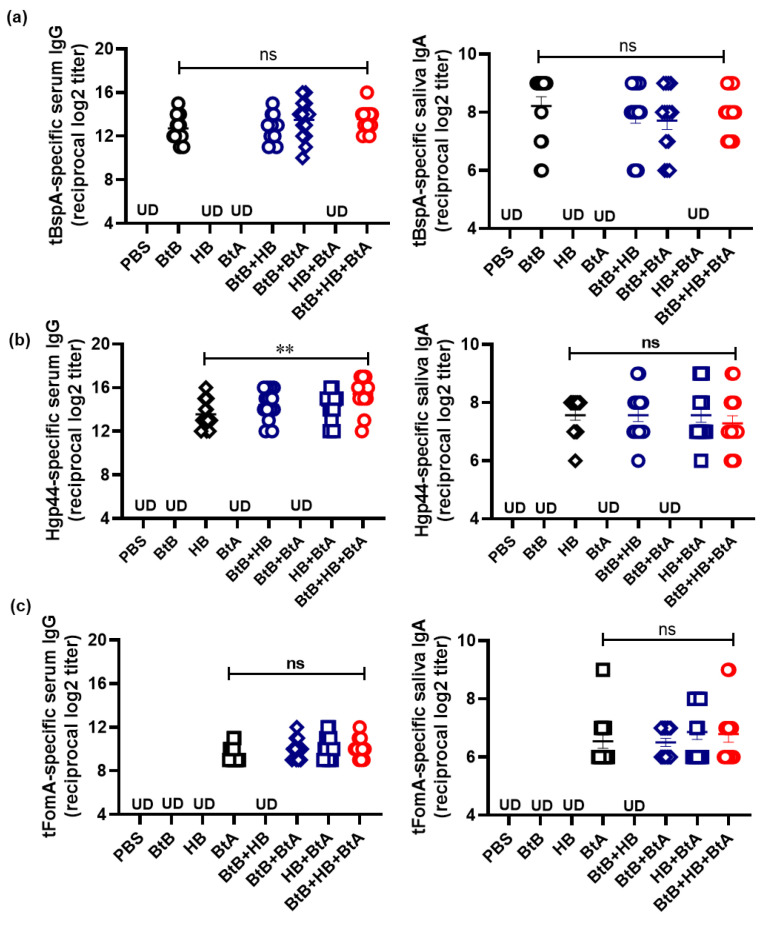
The trivalent vaccine elicits both systemic and mucosal antibody responses. Seven-week-old female BALB/c mice were intranasally immunized 3 times at 2-week intervals with the following formulations: PBS, a monovalent vaccine (6.2 μg of BtB, 8 μg of HB, or 5.1 μg of BtA), a divalent vaccine [6.2 μg of BtB plus 8 μg of HB (BtB + HB), 6.2 μg of BtB plus 5.1 μg of BtA (BtB + BtA), or 8 μg of HB plus 5.1 μg of BtA (HB + BtA)], and a trivalent vaccine [6.2 μg of BtB plus 8 μg of HB plus 5.1 μg of BtA (BtB + BtA + HB)]. (**a**–**c**) Serum and saliva samples were collected 2 weeks after the final immunization to measure tBspA-, Hgp44-, and tFomA-specific antibody responses using ELISA. The results are presented as the mean ± SEM for each group. *n* = 14; ** *p* < 0.01; ns indicates non-significance; UD indicates under the detection limit.

**Figure 3 vaccines-12-00754-f003:**
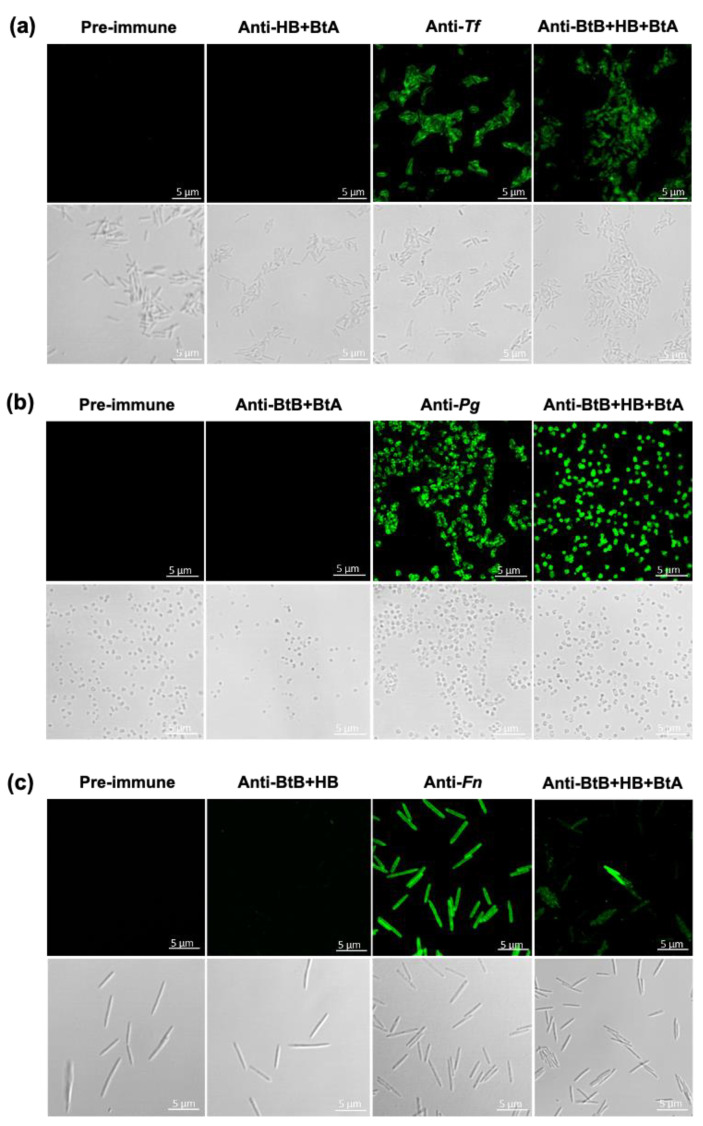
Anti-sera induced by the trivalent vaccine detects cognate antigens on the surfaces of live *Tf*, *Pg*, and *Fn*. Immunofluorescence was employed to detect the natural forms of BspA, Hgp44, and FomA expressed on the surfaces of live *Tf* (**a**), *Pg* (**b**), and *Fn* (**c**), respectively. Freshly cultured bacteria were incubated with anti-sera derived from naïve mice (pre-immune sera), mice vaccinated with *Tf* (anti-*Tf), Pg* (anti-*Pg*)*, Fn* (anti-*Fn*)*,* divalent (anti-HB + BtA, anti-BtB + BtA, or anti-BtB + HB), or trivalent (anti-BtB + HB + BtA) vaccines. The specimens were visualized using confocal microscopy.

**Figure 4 vaccines-12-00754-f004:**
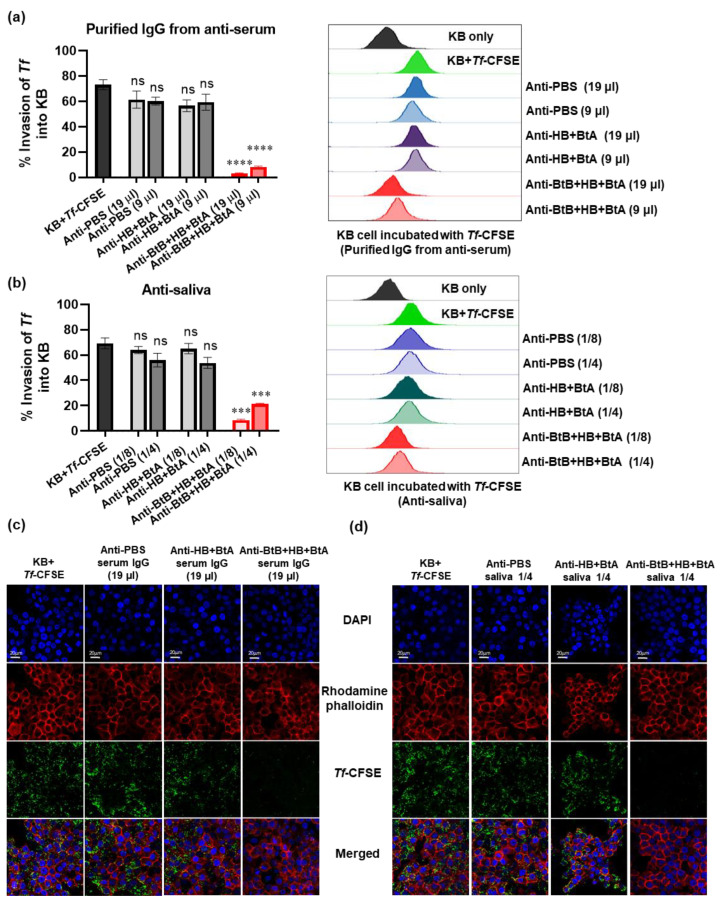
The anti-sera and anti-saliva raised by the trivalent vaccine inhibit KB cell adhesion and invasion by *Tf*. KB cell adhesion/invasion by *Tf* was determined by using flow cytometry and confocal microscopy. CFSE-labeled *Tf* was pre-incubated with IgG purified from anti-sera (equivalent to 19 μL/well or 9 μL/well) or anti-saliva (equivalent to 1/4 or 1/8 dilution) obtained from mice immunized with PBS (anti-PBS), a divalent vaccine (anti-HB + BtA), or a trivalent vaccine (anti-BtB + BtA + HB) for 1 h. Then, KB cells were infected with CFSE-labeled *Tf* at an MOI of 1:100 for 4 h. (**a**,**b**) Cells were analyzed by flow cytometry after quenching the fluorescence of bacteria bound to the surface with trypan blue. (**c**,**d**) Representative confocal microscopic images illustrating adhesion and invasion by *Tf* treated with purified IgG from anti-sera (equivalent to 19 μL/well) or anti-saliva (equivalent to 1/4 dilution), respectively. Data are represented as the mean ± SEM from three independent flow cytometry experiments. *** *p* < 0.001, **** *p* < 0.0001, and ns indicates non-significance.

**Figure 5 vaccines-12-00754-f005:**
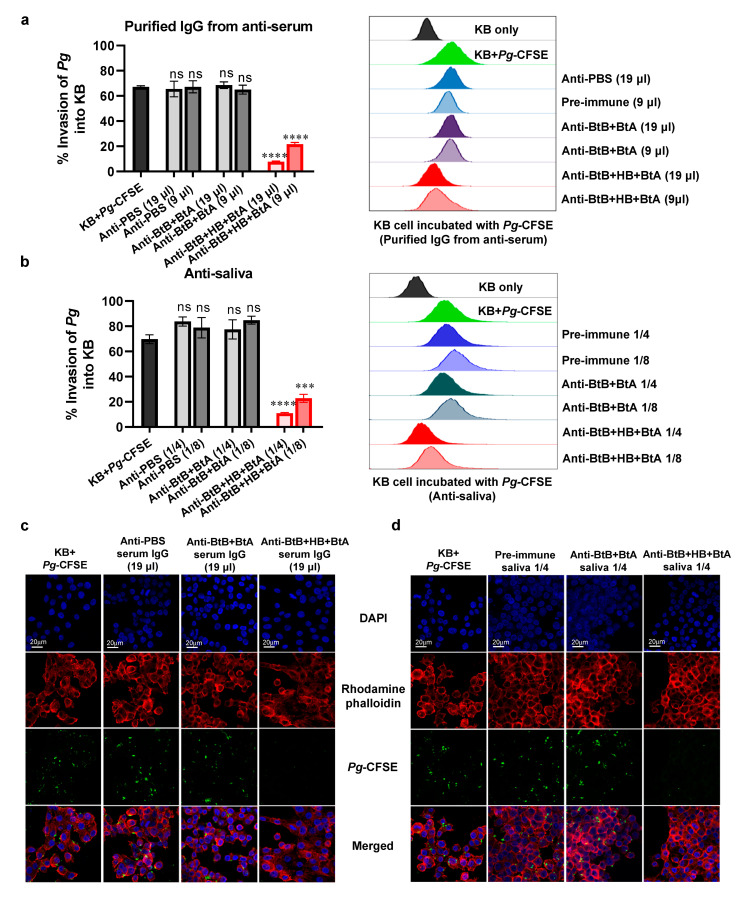
The trivalent vaccine-derived anti-sera and anti-saliva inhibit KB cell adhesion and invasion by *Pg*. The inhibition of KB adhesion/invasion by *Pg* was determined by flow cytometry and confocal microscopy. CFSE-labeled *Pg* was pre-incubated with IgG purified from anti-sera (equivalent to 19 μL/well or 9μL/well) or anti-saliva (equivalent to 1/4 or 1/8 dilution) derived from mice vaccinated with PBS (anti-PBS), a divalent vaccine (anti-BtB + BtA), or a trivalent vaccine (anti-BtB + BtA + HB) for 1 h. KB cells were infected with *Pg* at an MOI of 1:100 for 4 h. (**a**,**b**) Cells were analyzed by flow cytometry after quenching the fluorescence of bacteria bound to the surface with trypan blue. (**c**,**d**) Representative confocal microscopic images illustrating adhesion and invasion by *Tf* treated with purified IgG from anti-sera (equivalent to 19 μL/well) or anti-saliva (equivalent to 1/4 dilution), respectively. Data are represented as the mean ± SEM from three independent flow cytometry experiments. *** *p* < 0.001, **** *p* < 0.0001, and ns indicates non-significance.

**Figure 6 vaccines-12-00754-f006:**
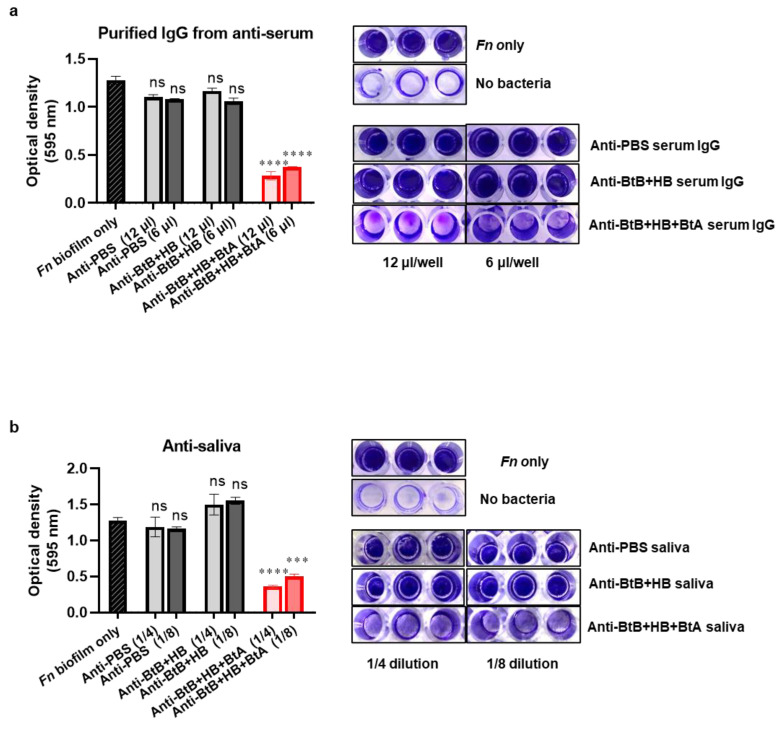
The anti-sera and anti-saliva produced by the trivalent vaccine prevent the formation of *Fn*-mediated biofilm. The inhibition of *Fn* biofilm formation. (**a**,**b**) *Fn* (1 × 10^9^ cells) was pre-incubated with IgG purified from anti-sera (equivalent to 12 μL/well or 6 μL/well) or anti-saliva (1/4 or 1/8) from mice vaccinated with PBS (anti-PBS), a divalent vaccine (anti-BtB + HB), or a trivalent vaccine (anti-BtB + BtA + HB) for 3 h at RT, followed by overnight incubation at 37 °C under anaerobic conditions (85% N_2_, 10% H_2_, and 5% CO_2_). Following a gentle wash with PBS, the wells were stained with 0.3% crystal violet for 15 min. The stained biofilm was extracted with 100% ethanol and diluted twofold with PBS, and the absorbance was measured at 595 nm. The right panels represent a microscopic observation of the crystal violet-stained biofilm. The presented data represent the mean ± SEM for each group, and the experiments were conducted with three replicates. *** *p* < 0.001, **** *p* < 0.0001, and ns indicates non-significance.

## Data Availability

The raw data supporting the conclusions of this article will be made available by the authors upon request.

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
