# Peer review of "A Flagellin-Adjuvanted Trivalent Mucosal Vaccine Targeting Key Periodontopathic Bacteria"

_vaccines, 2024, doi:10.3390/vaccines12070754_

Round 1

Reviewer 1 Report

Comments and Suggestions for Authors

Many Refs, related to "Materials", are missing

Tables should be more useful

State the model(s) used as sub-titles

No trade-marks in Scientific articles

Author Response

Many Refs, related to "Materials", are missing

  • Thank you for bringing this to our attention. Regarding the missing reference in the material and methods section, we have thoroughly reviewed it and included relevant references that guide the experiment protocol. We have also ensured that the necessary citations are now correctly incorporated.

Tables should be more useful

  • Thank you for your suggestion to present the data in table format. We appreciate your comment and agree that tables can concisely summarize the data. However, we believe that graphs allow for a more direct interpretation and explanation of the results of each experiment. Nonetheless, we are willing to provide detailed data in table format upon request to ensure comprehensive accessibility for all readers.

State the model(s) used as sub-titles

  • Thank you for your suggestion. We have addressed this by stating that the model “temporary ligature plus oral infection” was used as a sub-title in materials and methods section 2.3 and result section 3.1.

No trade-marks in Scientific articles

  • We appreciate your comment on the issue of trademarks in scientific articles. We fixed it.

Reviewer 2 Report

Comments and Suggestions for Authors

This manuscript was written to present significant findings on the effects of intranasal immunization for experimental periodontal disease. The study was quite complete and may interest to many readers. There are a couple of point the authors may need to revise.

1. It is recommended to remove the paragraph located on page 5, lines 283 to 285.

2. Correction of the Timeline on the X-Axis of animal experiments. The corrected timeline should be as follows:

  • Intranasal Immunization Period:
    • Start: Day -36
    • End: Day -22 (assuming the duration is 14 days as depicted from -14 to 0 in the original image)

Author Response

This manuscript was written to present significant findings on the effects of intranasal immunization for experimental periodontal disease. The study was quite complete and may interest to many readers. There are a couple of point the authors may need to revise.

  1. It is recommended to remove the paragraph located on page 5, lines 283 to 285.
  • We appreciate your bringing this to our attention. Following your comment, we have removed these irrelevant sentences.

  1. Correction of the Timeline on the X-Axis of animal experiments. The corrected timeline should be as follows:

Intranasal Immunization Period:

    1. Start: Day -36
    2. End: Day -22 (assuming the duration is 14 days as depicted from -14 to 0 in the original image)

  • Thank you for your valuable comment. We have revised the experimental design timeline, as shown in the attached photo, to improve its clarity. Here are the details of the updated timeline: the experiment was divided into two continuous parts.
  • Mice were immunized three times at 14-day intervals (-42, -28, -14).
  • Fourteen days after the third immunization (-14 - 0), we began the temporary ligature model corporate with an oral bacterial infection, which lasted 15 days.
  • We treated mice with antibiotic/free antibiotics in drinking water for six days before starting the ligation placement.  

Reviewer 3 Report

Comments and Suggestions for Authors

Author Response

Dear Author Congratulation for your effort Please In the abstract direct better the last sentence as follow: In summary, the flagellin-adjuvanted trivalent mucosal vaccine-mediated immunomodulation would be a promising choice for clinically preventing dysbiotic bacteria-induced periodontitis. Clinical studies are needed to clarify its role in managing of periodontitis patients.

  • Thank you for the suggestions. Following your guidance, we added the summary in the abstract.

Reviewer 4 Report

Comments and Suggestions for Authors

The authors conducted a very interesting study;

An experimental system was established using a temporary ligation and oral infection with live bacteria to simulate periodontal disease (PD) and evaluate the efficacy of the trivalent mucosal vaccine.

trivalent mucosal vaccine was developed that targets the surface virulence factors of Tf (BspA), Pg (Hgp44), and Fn (FomA), fused to the mucosal adjuvant FlaB. Intranasal immunization with the trivalent vaccine prevented alveolar bone loss and the production of proinflammatory cytokines in the experimental model.

The vaccinated mice exhibited significant induction of Tf-tBspA-, Pg-Hgp44-, and Fn-tFomA-specific IgG and IgA responses in the serum and saliva, respectively. The anti-sera and anti-saliva efficiently inhibited epithelial cell invasion by Tf and Pg and interfered with biofilm formation by Fn. In summary, flagellin-adjuvanted trivalent mucosal vaccine-mediated immunomodulation would be a promising option for clinically managing dysbiotic bacteria-induced periodontitis.

Animal model: Differences between the human and mouse oral microbiome may limit the extrapolation of results. This is a preclinical study, the safety of the vaccine in humans has not yet been evaluated Study duration: The long-term effects of the vaccine were not evaluated. specific response: The vaccine targets three specific bacteria, which may not cover all variations of dysbiosis

 The Materials and Methods section of the study is well structured and covers the essential aspects Details: Ethical approvals and animal care following university guidelines and relevant legislation are mentioned. The description is clear and provides detailed steps to reproduce the model. Intranasal immunization procedures are detailed, including doses of vaccine components and time intervals between immunizations. Providing additional details about the negative and positive controls used in the experiments could increase the validity and reproducibility of the results.

Integrate  references to recent and relevant studies on the treatment of microbial dysbiosis and the development of vaccines for periodontal diseases.

Author Response

The authors conducted a very interesting study.

An experimental system was established using a temporary ligation and oral infection with live bacteria to simulate periodontal disease (PD) and evaluate the efficacy of the trivalent mucosal vaccine.

trivalent mucosal vaccine was developed that targets the surface virulence factors of Tf (BspA), Pg (Hgp44), and Fn (FomA), fused to the mucosal adjuvant FlaB. Intranasal immunization with the trivalent vaccine prevented alveolar bone loss and the production of proinflammatory cytokines in the experimental model.

The vaccinated mice exhibited significant induction of Tf-tBspA-, Pg-Hgp44-, and Fn-tFomA-specific IgG and IgA responses in the serum and saliva, respectively. The anti-sera and anti-saliva efficiently inhibited epithelial cell invasion by Tf and Pg and interfered with biofilm formation by Fn. In summary, flagellin-adjuvanted trivalent mucosal vaccine-mediated immunomodulation would be a promising option for clinically managing dysbiotic bacteria-induced periodontitis.

Animal model: Differences between the human and mouse oral microbiome may limit the extrapolation of results. This is a preclinical study; the safety of the vaccine in humans has not yet been evaluated

  • Thank you for your insightful concern. We acknowledge the importance of your comment and recognize that assessing the vaccine's safety in humans is a crucial aspect that requires further investigation. As our study primarily focuses on establishing proof-of-concept regarding the effectiveness of the trivalent mucosal vaccine against dysbiotic bacteria-induced periodontitis in mice, the safety profile of this vaccine in humans has not yet been evaluated. This limitation underscores the preliminary of our research and highlights the need for further investigation, including safety assessments in clinical trials. Our study's “temporary ligature plus oral infection” mouse model provides a fundamental platform for temporary colonization of human periodontal-induced PD, which can effectively demonstrate the vaccine's efficacy.

Study duration: The long-term effects of the vaccine were not evaluated.

  • Thank you for raising this critical concern. Our study focused primarily on short-term outcomes following intranasal immunization with the vaccine against forsythia, P. gingivalis, and F. nucleatum, demonstrating significant reductions in alveolar bone loss and proinflammatory cytokine production in a mouse model of periodontal disease. While these initial results are promising, we recognize the importance of conducting longitudinal studies to assess the vaccine's long-term impact on periodontal health. Future investigations will evaluate the durability of immune responses and the potential for disease recurrence prevention.

Specific response: The vaccine targets three specific bacteria, which may not cover all variations of dysbiosis​

  • Thank you for your insightful concern regarding our study. We understand that targeting only three specific bacteria may not cover all variations of dysbiosis. However, our study targets the specific virulence factors of Tf-BspA, Pg-Hgp44, and Fn-tFomA due to their crucial roles in the pathogenesis of dysbiotic periodontal microbiota. Fusobacterium nucleatum serves as a physical bridge between early and late colonizers of dental plaque organisms, facilitating biofilm maturation. Porphyromonas gingivalis, as a keystone pathogen, deregulates the immune response, leading to dysbiosis. Tannerella forsythia is a member of the red-complex bacteria, which is more invasive and interacts with other organisms. We believe that our vaccine approach, which targets the key virulence factors of these keystone bacterial pathogens, will not only reduce the levels of these pathogens but also modulate the dysbiotic status of oral bacteria, thereby delaying the progression of periodontal disease. Future studies could explore the inclusion of additional bacterial components to enhance the vaccine's coverage against a broader spectrum of dysbiosis observed in clinical scenarios. We hope this explanation addresses your concern and underscores the scientific basis of our approach.

The Materials and Methods section of the study is well structured and covers the essential aspects Details: Ethical approvals and animal care following university guidelines and relevant legislation are mentioned. The description is clear and provides detailed steps to reproduce the model. Intranasal immunization procedures are detailed, including doses of vaccine components and time intervals between immunizations. Providing additional details about the negative and positive controls used in the experiments could increase the validity and reproducibility of the results.

  • Thank you for your insightful comment. To enhance the validity and reproducibility of experimental results, we have included both positive and negative controls. These controls are detailed in materials and methods, as well as in results. In confocal and adhesion/invasion assay, we used serum-IgG/ saliva from mice vaccinated with PBS as negative control and generated anti-Tf, anti-Pg, and anti-Fn with inactivated whole-cell bacteria as positive control. Additionally, we included another negative control anti-serum/anti-saliva derived from a non-specific immune response.

Integrate references to recent and relevant studies on the treatment of microbial dysbiosis and the development of vaccines for periodontal diseases.

  • Thank for your comment. We have updated our manuscript to include relevant studies on the treatment of microbial dysbiosis and the development of vaccines for periodontal diseases.

Reviewer 5 Report

Comments and Suggestions for Authors

The aim of the present study was to develop a recombinant mucosal vaccine targeting a surface-associated protein 26 BspA of Tf by genetically combining truncated BspA with built-in adjuvant flagellin (FlaB).

The topic of this reaserch is a very interesting one.

The study was well designed and the idea is very inovative.

Regarding figure 1 there are too many informations in it.

Please devide it in at least 2 pictures.

In the discussion section please include more recent published articles.

The present study does not include also a conclusion.

Please include at the end of the discussion chapter a conclusion.

Comments on the Quality of English Language

Moderate

Author Response

The aim of the present study was to develop a recombinant mucosal vaccine targeting a surface-associated protein 26 BspA of Tf by genetically combining truncated BspA with built-in adjuvant flagellin (FlaB).

The topic of this reaserch is a very interesting one.

The study was well designed and the idea is very inovative.

Regarding figure 1 there are too many informations in it.

Please divide it in at least 2 pictures.

In the discussion section, please include more recent published articles.

The present study does not include also a conclusion.

Please include a conclusion at the end of the discussion chapter.

  • We appreciate your comments regarding Figure 1. We have consolidated the in vivo animal model data into a single figure to provide a clear and consistent presentation of our experimental results. This figure demonstrated the efficacy of the mucosal vaccine within the experiment's timeline, showing protection against bone resorption and subsequent periodontal gene expression. This comprehensive approach aims to enhance understanding by presenting all relevant data in one cohesive figure.